# Evaluation and Improvement of Greenness for Milling AL6061 Alloy through Life Cycle Assessment and Grey Relational Analysis

**DOI:** 10.3390/ma15228231

**Published:** 2022-11-19

**Authors:** Zhipeng Xing, Haicong Dai, Jiong Zhang, Yufeng Li

**Affiliations:** 1Institute for Advanced Materials and Technology, University of Science and Technology Beijing, Beijing 100083, China; 2Shanghai Aerospace Equipments Manufacturer Co., Ltd., Shanghai 200245, China; 3State Key Laboratory of Mechanical Transmission, Chongqing University, Chongqing 400044, China

**Keywords:** grey relational analysis, life cycle assessment, milling process, environmental impact, surface quality

## Abstract

Modern manufacturing industries thrive on greenness, which means ensuring acceptable environmental impacts and required surface quality of the products during the manufacturing process. However, there is a conflict between surface quality and environmental performances in the milling process. The current research only considers energy consumption rather than total environmental impacts. In this respect, this research focuses on the multiobjective optimization of machining parameters for balancing the surface quality (i.e., surface roughness, Ra) and total environmental impact (TEI), which includes raw materials usage, energy consumption, and output pollutant emission during the milling of AL6061 alloy. First, life cycle assessment (LCA) of the milling process is used for evaluating the TEI. Then, multiobjective optimization is conducted using Grey Relational Analysis. The results indicated that the improvement of Ra and TEI can be achieved with higher cutting speed, higher depth, and wet conditions in milling. The optimization work showed that cutting speed of 165 m/min, feed rate of 0.28 mm/rev, depth of cut of 2 mm, and width of cut of 3 mm are the optimal combination among existing experiments. Compared to single objective optimization results, multiple responses (Ra and TEI) can be improved simultaneously.

## 1. Introduction

The manufacturing industry has played an essential role in the economic development of each country in the world [1]. Due to the National Association of Manufacturers, the largest and most influential employer association in the United States in the 1930s and 1940s, the manufacturing industry alone accounts for about 65% of the energy consumption of the industrial sector, and is expected to continue growing in the coming decades [2]. The manufacturing industry also produces significant amounts of greenhouse gases, accounting for 19% of the total greenhouse gas emissions from various economic activities [3]. The Chinese government has made greenhouse emissions control a major national strategy [4]. Green manufacturing, which is a process or system which has a minimal, nonexistent, or negative impact on the environment [5], is a vital link for achieving emissions reduction, and its aim is to improve machining general performance while reducing raw materials usage, energy consumption, and output pollutant emission, thus reducing negative environmental impacts [6].

Machining is one of the fundamental manufacturing technologies, and its material-removal characteristics inherently render it wasteful in its use of energy [7]. Milling is the process in which a surface is generated by the progressive formation and removal of chips of materials from a workpiece as it is fed to a rotating cutter in a direction perpendicular to the axis of the cutter. Moreover, milling operation accounts for a significant proportion of the total environmental impact (TEI) of certain products, such as molds and dies [8]. Hence, the milling process has a highly green potential for saving energy, as well as for reducing gas emissions and the production of waste materials [9].

Some studies have been conducted to analyze the quality of machined parts during the milling processes. The quality assessment of machined parts depends on different functional aspects. One aspect used to evaluate the milling quality, shared in all types of parts, is the surface finishing or roughness. This parameter will be used in this study as the reference machining quality parameter due to its general application. Wang et al. [10] analyzed the influence of cutting condition (cutting speed, feed, depth of cut, et al.) on surface roughness when milling AL2014-T6. Wang et al. [11] constructed a milling surface roughness prediction model considering the mechanical factors (cutting speed, feed rate, cutting depth, and cutting width) that influence surface roughness during milling Ti6Al4V.

For milling, considering the machining quality but ignoring environment performance is not possible in the new production context of green manufacturing. Three hotspots in environmental performance exist in the application of milling. The first one is the high energy and resource consumption. The cutting process consumes significant energy, which has a negative effect on the environment. This is because energy consumption is linked with environmental pollution and resource depletion. Almost all energy-generating processes pollute the environment in some way [12], and continuously increasing demand has made energy a scarce resource [13]. For energy efficiency improvement and energy saving in milling processes, much research has been done; for example, some methods include energy modeling and evaluation, as well as energy optimization for machining processes and machines [14]. Second, cutting fluids are often used in cutting processes to improve machining performance by reducing the friction or temperature of the cutting region. However, the application of cutting fluid causes great harm to human health and the environment due to the generation of toxic and hazardous wastes [15]. Third, the generation of airborne particulates or aerosols during emission is very common in machining processes. Aerosols are defined as fine or ultrafine solid particles from metals or alloys (metal dust) or liquid particles from cutting fluids (oil mists) [4]. They could cause serious environmental problems and are detrimental to the human respiratory system [16]. Green manufacturing strategies thus require researchers to provide a methodology for evaluating the environment impacts of resources, energy consumption, and pollutant emission comprehensively.

In addition, it is essential to evaluate the environmental performance of the milling process. Input-process-output model (IPO) analysis can help to better understand the environmental performance, i.e., resources, energy consumption, and pollutant emission, in the process of parts machining. Jiang et al. [17], Bourhis et al. [18], and Lv et al. [19] used the IPO model to analyze the effect of the manufacturing process on the environment. However, it is difficult to understand or compare the environmental impacts between different milling processes solely based on separate IPO analysis without considering the materials flow. Therefore, a full environmental impact analysis is required in order to explore more general solutions for environmental assessment and optimization.

Life cycle assessment (LCA) is a preferred and widely used method for a full environmental impact analysis of a process or product and has been standardized in both ISO 14040 and ISO 14044 [20]. In addition, it enables materials flow to be converted into a range of environmental impact categories, so that comparisons can be made among different milling processes. Some previous studies on LCA with process parameter selection were published [21,22,23]. Nevertheless, they did not discuss the improvement of environmental performance through optimizing the milling machining process parameters.

Some researchers focused on improving the environment performances through optimizing machining process parameters or cooling technologies during the milling processes. However, the requirement of reducing environmental impacts and the practices of optimizing the milling process may lead to a partial sacrifice of machining quality due to the existence of conflicting aspects. Camposeco-Negrete [24] analyzed the influence of process parameters on responses of AISI 6061 T6 aluminum in order to obtain its minimum energy consumption and maximum cutting quality. The analysis results showed that higher feed rate provided minimum energy consumption but led to higher surface roughness. Du et al. [25] analyzed and optimized the energy consumption and surface quality during dry machining of 304 stainless steel, and clarified the contradiction between energy consumption and surface quality. Ming et al. [26] confirmed that energy efficiency and exhaust emissions (based on the indexes of Particulate Matter PM1.0, PM2.5, and PM10) are negatively correlated with surface quality for both Al 6061 and SKD 11 materials during electrical discharge machining. Vukelic et al. [27] evaluated and optimized machining parameters from the aspects of surface quality and environmental impact in the turning of Inconel 601. However, the study focused on optimizing the environmental impact by adjusting the cutting parameters, without considering the impacts contributed by cutting fluid consumption. This renders it difficult to provide sufficient information to make improvements to the milling process, such as selecting cooling technologies.

Therefore, to overcome the problem, the milling process is investigated with LCA and grey relational analysis (GRA) to balance out the environmental impacts and machining quality, which is referred to as green-high performance of the milling process in this work. The rest of this paper is structured as follows: Section 2 introduces the research methodology. Section 3 introduces the experimental procedure. Section 4 introduces the optimization method using the GRA. Section 5 introduces the results and discussion.

## 2. Research Methodology

The research methodology for this study is shown in Figure 1. Firstly, the ranges of the machining parameters are defined according to the characteristics of the workpiece and recommendations. In this manner, machining parameters such as cutting speed *v_c_*, feed rate *f*, milling depth *a_p_*, milling width *a_e_,* and cooling technologies (CT) of milling are obtained. Then, the milling experiments are conducted on selected machine tools with adequate fixtures and milling tools. In order to render the LCA able to evaluate and analyze the negative impacts on the environment, a selected list of environmental-related parameters are measured, including parameters such as processing time, power, dust, noise, and gaseous pollutant (oil mist, carbon monoxide, nitrogen dioxide, ozone, and sulfur dioxide). After the experiment, the surface roughness and waste cutting fluid are measured. Then, the above collected data are used for LCA, which is used to evaluate and analyze the negative impacts on the environment. Finally, a multiobjective optimization method based on GRA [28] is proposed to determine an optimal set of input parameters. Optimal condition refers to minimizing the surface roughness (SR) and the total environmental impacts (TEI), both depending on the machining parameters as depicted in Equation (1):(1)$$\left\{\begin{array}{c} \min\mathrm{SR}\left(v_{c},f,a_{p},a_{e},\mathrm{CT}\right)\\ \min\mathrm{TEI}\left(v_{c},f,a_{p},a_{e},\mathrm{CT}\right) \end{array}\right.$$
where *v_c_* is the cutting speed; *f* is the feed rate, *a_p_* is the milling depth, *a_e_* is the milling width, and CT presents the CT.

## 3. Experimental Procedures

### 3.1. Workpiece and Cutting Tools

Aluminum AL6061 is a common alloy with many purposes, since it has superior mechanical properties such as hardness and weldability. It is commonly used in the aircraft, automotive, and food packaging industries [29]. Therefore, AL6061 alloy is selected as the material of a workpiece for performance in a CNC milling machine. The original workpiece size is 50 × 50 × 20 mm^3^ and one flat surface was performed, considering the limited time and reasonable conditions to guarantee the quality of the measurements. The experiment was performed on a CNC milling machine model VMCL850 (Tontec International Ltd., Kowloon Bay, Hong Kong, China) to mill a flat surface. The technical details of this machine are as follows: position accuracy of 0.01 mm; maximum spindle speed of 8000 rpm; workbench area is 510 mm × 1080 mm; power is 18 kW. Further, according to the workpiece material and the manufacturers’ recommendations, the carbide-coated end mill with four unequal spiral edges was selected as the cutting tool. The experimental setup, as well as important characterization units, are shown below in Figure 2.

### 3.2. Design of the Experiment

Several lubrication strategies, i.e., dry machining, wet machining, and cryogenic cooling [30], etc., have been implemented in different machining operations. Two cooling and lubrication technologies were used in this study for the milling process: dry condition, with neither external cooling nor lubrication, and wet condition, adding cutting fluid ECOCOOL 1030 S (FUCHS PETROLUB SE, Mannheim, Germany) for cooling and lubrication with a speed of 0.68 g/min by the automatic oil pump controlled by the CNC system of the milling machine.

The optimization was performed in terms of the mentioned input machine parameters (*v_c_*, *f*, *a_p_*, *a_e_*, CT) according to the Equation (1). The machining parameter ranges can be defined based on the characteristics of machine tools, empirical data, the literature recommendations, research papers, and tool manufacturer’s recommendations. In this case, each range was spilt into four points: cutting speed, *v_c_* (90, 115, 140, 165 m/min); feed rate, *f* (0.2, 0.24, 0.28, 0.32 mm/rev); milling depth, *a_p_* (0.8, 1.2, 1.6, 2 mm); milling width, *a_e_* (2.5, 3, 3.5, 4 mm). The cooling is defined as dry or wet. The experimental combinations of the milling parameters were obtained using the orthogonal array matrix in the Taguchi design of experiments [31], as shown in Table 1.

### 3.3. Measurement Equipment

Once the input parameters and their ranges were selected, experiments were performed using milling. During the experiments outlined in Figure 2, data related to environment impact were obtained by a series of sensors, which are shown in Figure 3. The power meter (PW3337-02, made by Hioki Electric Co., Ltd., Ueda, Japan, Range 3 W–150 kW) and its connection wire, shown in Figure 3a, were used for measuring energy consumption. In any machining environment, noise from machining tools is among the most common and frequent occupational hazards. A sound sensor (Suzhou TASI Electronic Industrial Co., Ltd., Suzhou, Jiangsu, China, Rage 30–130 dBA) was used for measuring noise data. A dust sensor (Hunan Rike Instrument Co., Ltd., Changsha, Hunan, China, PM2.5: 0.0–20 mg/m^3^ PM10: 0.0–50 mg/m^3^ PM100: 0.0–100 mg/m^3^) was used for measuring dust concentration. An air detector instrument (Hunan Rike Instrument Co., Ltd., Changsha, Hunan, China, Resolution 0.001 ppm) was used for measuring several kinds of gas which integrated with carbon monoxide, nitrogen dioxide, ozone, sulfur dioxide, and oil mist sensors, as shown in Figure 3b. The instrument for measuring Ra value MFT-5000 (Rtec instruments, Inc., San Jose, CA, USA) used the so-called white light interferometry technology to define the surface roughness, as shown in Figure 3c. Three points on the workpiece surface, which were obtained under each working condition, were randomly selected for measurement, and the average value was calculated.

### 3.4. Environmental Impact Assessment

LCA is used to evaluate the inputs, outputs, and potential environmental impacts of a product system throughout its life cycle [32]. The quantitative results of environmentally negative impacts are acquired by LCA and through subsequent optimization strategy are reduced. The LCA process is mainly divided into four parts.

#### 3.4.1. Goal and Scope Definition

A clear analysis objective and an accurate evaluation scope determine the direction and depth of analysis or evaluation. Therefore, it is essential to determine the system boundary of the milling process, which excludes those parts of the manufacturing system which are unnecessary or irrelevant to the analysis of environmental impacts. The system boundaries encompassed one life cycle stage (the milling phase) with two activities, i.e., material usage and energy consumption. The tool geometry, material, vibration during machining, and other factors were fixed under assumption throughout the experiments so that these factors needed not be considered.

#### 3.4.2. Inventory Analysis

Inventory analysis is the process for quantifying the use of resources and energy and the discharge of waste to the environment during the life cycle of the products. In this section, the resource usage, energy usage, and environmental release within the system boundary of the metal milling process were quantified and evaluated. The functional unit was defined as CNC face milling of AL6061 in Section 3.2.

In order to implement LCA, it is necessary to analyze the raw materials usage (including the AL6061 and cutting fluid), energy consumption, and output (including the noise, oil mist, dust, etc.,) during each experiment quantitatively. In this step, the input of materials and the energy consumption, as well as the emissions of various gases, are all counted and converted into units supported by Gabi software (V8.0) developed by THINKSTEP AG. This software is an environmental impact analysis software designed according to LCA methodology principles, and the content, transparency, and flexibility of its graphical interface are characteristics of the world’s largest dataset. It provides methods, explanations, and sensitivity analysis for systematic evaluation or distribution evaluation according to each project stage of life cycle assessment and life cycle engineering. In the process of using Gabi software, it is assumed that the cutting chips produced during the milling process can be recycled according to the actual processing status, and since the machine tool is mainly driven by electric energy, the energy consumption can be represented by the electricity production.

#### 3.4.3. Life Cycle Impact Assessment

In this phase, the output of milling process was classified and characterized at first, and Gabi was used to quantify and analyze the environmental impacts of the milling process. PETW_EU2004_ expresses the environmental impact from an average citizen according to the political targets for each of the impact categories in the year 2004. The index expresses that the reference region for the weighting and the underlying normalization is the EU and the target year used for the weighting is 2004 [33]. The index was selected as a collection of environmental impact quantification schemes, which included seven midpoint impact categories: acidification potential (AP), aquatic eutrophication (AE), global warming (GW), photochemical ozone formation—impact on human health and materials (POFH), photochemical ozone formation—impact on vegetation (POFV), stratospheric ozone depletion (SOD), and terrestrial eutrophication (TE).

## 4. The Optimization Method Using GRA

In this study, two objectives were considered: minimizing the SR and the TEI functions. GRA is a multiobjective response optimization technique developed by Deng in 1989 [28]. GRA techniques are commonly used for calculating the performance of complex optimization problems with wide variability and insufficient data. The GRA method evaluates different input sequences with the help of the so-called grey relational grade (GRG) parameter, which allows for the comparison. If two different input sequences result in very close results, the value of GRG is close to 1, and ideally it is equal to 1 if identical. The GRA method was applied in several steps, described below.

Step 1: Data Preprocessing

The original data use different units. To avoid the effect of response-variable units and minimize the variability, the original data were normalized into dimensionless indexes between the range of 0 and 1 [34]. According to Equation (1), the SR and TEI were considered as lower-the-better performance characteristics. Therefore, Equation (2) was used to normalize the data in a single dimension (0–1):(2)$$y_{i}\left(k\right)=\frac{\max x_{i}\left(k\right)-x_{i}\left(k\right)}{\max x_{i}\left(k\right)-\min x_{i}\left(k\right)}$$
where, $x_{i}\left(k\right)$ indicates the original experimental value of the *k*th response in the *i*th experiment, $y_{i}\left(k\right)$ represents the normalized data for the *k*th response in the *i*th experiment, and $\max x_{i}\left(k\right)$ and $\min x_{i}\left(k\right)$ are the maximum and minimum values of the original experimental data for the *k*th response, respectively. The highest value of the normalized data was represented as a superior performance characteristic and it was equal to one.

Step 2: Grey Relational Coefficient

After the normalization, the grey relational coefficient (GRC) was calculated to determine the relationship between the optimal and the actual normalized results. The GRC was expressed as in Equation (3) [35]:(3)$$v_{i}\left(k\right)=\frac{\Delta_{\min\;}+\zeta \Delta_{\max}}{\Delta_{0i}\left(k\right)+\zeta \Delta_{\max}}$$
where, $\Delta_{0i}\left(k\right)$ indicates the deviation sequence; this deviation sequence can be presented as Equations (4)–(6):(4)$$\Delta_{0i}\left(k\right)=\left|y_{0}\left(k\right)-y_{i}\left(k\right)\right|$$
(5)$$\Delta_{\min}=\underset{\forall i}{\min\;}\underset{\forall k}{\min\;}\Delta_{0i}\left(k\right)$$
(6)$$\Delta_{\max}=\underset{\forall i}{\max\;}\underset{\forall k}{\max\;}\Delta_{0i}\left(k\right)$$

In the above equation, the ζ is termed as the distinguishing coefficient, and it can have any value between 0–1. In this study, ζ is taken as 0.5 to fit the practical requirement, according to the study in [36]. Its function is to rectify the difference among relation coefficients.

Step 3: Grey Relational Grade

In this stage, a combined factor called GRG was calculated by combining the GRC with the associated weight value. The weight value $w_{k}$ was calculated by the principal component analysis, which is a useful statistical method to convert multiple indicators into several compositive ones [37]. GRG, represented as $\xi \left(y_{0},y_{i}\right)$, was calculated using Equation (7) [36], which is also defined as the greenness in this study:(7)$$\xi \left(y_{0},y_{i}\right)=\overset{n}{\underset{k=1}{\sum }}\omega_{k}v_{i}\left(k\right)$$

## 5. Results and Discussion

### 5.1. TEI Assessment by LCA

The LCA values were evaluated (see Section 3.4) and the results are shown in the charts of Figure 4 for each of the impact categories (AP, AE, GW, POFH, POFV, SOD, and TE; see Section 3.4.3); each are shown according to the different experimental conditions selected in Table 1. Considering the results of GW in Figure 4, the experiment No. 23 has the maximum value of GW with 4.24 × 10^−4^ and the experiment No. 12 has the minimum value of GW with 3.60 × 10^−4^. The main reason for this is that the experiment No. 23 has the smaller feed rate and smaller width of cut compared to the experiment No. 12, which results in more energy consumption and a further GW. Similar to GW, the impact categories AP and AE in experiment No. 23 are the maximum and those in experiment No. 12 are the minimum.

Further, the percentage contribution of the impact categories on TEI is shown in Figure 5. In all experiments, the percentage values of the impact categories of GW range between 40.57% to 40.90% of the total environmental impact. As the same comparison yields, the percentages for POFV, POFP, and AP are 14.66–15.01%, 14.36–14.71%, and 13.56–14.30%, respectively. The rest of the impact categories, including AE, TE, and SOD, are less than 10%. Consequently, the contribution of impact category GW to TEI is significantly larger than the others. The smallest contribution is the impact category SOD.

### 5.2. Calculated GRC and GRG

A higher weighted GRG value means better performance characteristics of the responses of the SR and TEI functions. Based on the GRA method (Section 4), the weight GRG is calculated by the Matlab 2022a software, and the rank of each experiment, according to the weight-GRG value, as presented in Table 2. The experiment No. 16 is ranked first place as having the maximum weighted-GRG value; therefore, its ‘greenness’ quality is the highest among all of the experiments. Hence, the experiment No. 16 is the best experimental setting, while the lowest one is No. 28. The machining setting respective to No. 16 indicates the optimum run. In other words, the optimal parameters are a cutting speed of 165 m/min, a feed rate of 0. 28 mm/rev, a milling depth of 2 mm, and a milling width of 3 mm under the wet cooling conditions.

### 5.3. Effect of Machining Parameters on the SR and TEI

In order to identify the significant effect of each machining parameter (milling speed, feed rate, milling depth, milling width, and CT) on the surface roughness (Ra), the main effect analysis results are shown in Figure 6 based on the experimental results. In this figure, the x-axis shows the values of each factor at different levels, and the y-axis shows the GRC value of SR. The maximum value represents the best response. It can be observed that the main effects are cooling technology and feed rate, while the milling speed, milling depth, and milling width seem to have a slightly lower effect. Taking the feed rate as an example, the GRC value of SR increases from 0.2 mm/rev to 0.24 mm/rev and decreases from 0.24 mm/rev to 0.32 mm/rev. Hence, when selecting the feed rate, the optimal Ra values of the feed rate are set as 0.24 mm/rev. For the cooling technology, the SR with the wet cooling condition is better than the SR with the dry cooling condition in the experiment. The use of cutting fluid would provide lubrication and cooling effects during the machining process, which improve the SR of the workpiece. Similarly, Zubira et al. [15] also reported that the preferred condition is wet machining rather than dry machining.

Further, the significant effects of each of the machining parameters on the TEI are analyzed, as shown in Figure 7. The results revealed that the significant factors are milling depth and cooling technology. For the milling depth, the change of milling depth can reduce the machining time and the energy consumption so that the negative environmental impact can be reduced correspondingly. The effect of cutting conditions on environmentally negative impacts is large. With the application of cutting fluid, the adverse health effects and environmental pollution would cause the generation of toxic and hazardous wastes. In the situation of dry cutting, more energy was consumed compared to wet cutting.

### 5.4. Multiobjective Optimization by GRA

According to the main effect analyses in Figure 4 and Figure 5, the optimal Ra values of the feed rate are set as 0.24 mm/rev, while the optimal TEI values of the feed rate are set as 0.32 mm/rev. As the feed increases, machining time is reduced, and environmental impact is reduced while surface roughness is increased. In terms of milling width, the optimal Ra values of the milling width are set as 3 mm, while the optimal TEI values of the milling width are set as 4 mm. Therefore, there is a conflicting aspect between the machining quality and the environmental impacts in the machining process.

Based on the results, the machining parameters can have a positive or negative impact on the GRC response, and the magnitude of the effect may be quite different. In order to highlight the differences of multiobjective optimization, we compared three milling strategies which are focused on obtaining an optimal surface roughness, an optimal environment negative impacts, and an optimal multiobjective result.

The result of the optimal strategy with regard to SR shows that machining should be performed using higher milling speed, lower feeds, higher depths of mill, lower milling width, and wet cooling conditions. For an optimal SR, the GRC for TEI in experiment No. 12 is the largest, as shown in Table 2. Hence, experiment No.12 is the optimum strategy regarding the TEI, in which cutting speed is 140 m/min, feed rate is 0.32 mm/rev, milling depth is 0.8 mm, milling width is 3 mm, and wet cooling conditions are used. The result of the optimal strategy with regard to the environmental impact shows that machining should be performed using a higher milling speed, higher feeds, higher depths of mill, a higher milling width, and wet cooling conditions. For as few environmentally negative impacts as possible, the GRC for SR in experiment No. 9 is the largest, as shown in Table 2. Therefore, experiment No. 9 is the optimum strategy with regard to environmental impact, in which cutting speed is 165 m/min, feed rate is 0.24 mm/rev, milling depth is 0.8 mm, milling width is 3.5 mm, and wet cooling conditions are used. Finally, during the multiobjective optimization, the experiment No. 16 is the optimum run, with the largest GRG as shown in Table 2.

The comparison results among the three milling strategies from the SR and TEI are shown in Figure 8. The milling strategy for an optimal SR has the minimal SR but the largest TEI, while the milling strategy for an optimal environment negative impact has the minimal TEI but the largest SR. Moreover, the values of SR and TEI for an optimal multiobjective result are in between these values for an optimal surface roughness strategy and an optimal environment negative impact strategy. Comparing the SRs of the three strategies, it can be observed that increasing the cutting speed and decreasing the feed rate can decrease SR. Comparing the TEIs of the three strategies, it is observed that increasing the feed rate and decreasing the cutting speed can decrease TEI, which is the opposite to the case of SR. It is clearly demonstrated that the multiple responses in the cutting process could be improved together by using the proposed optimization method.

## 6. Conclusions

Modern manufacturing industries strive for greenness, which means acceptable environmental impact criteria in place and a required surface quality of the workpiece. Milling machining parameters and CT were evaluated from the aspects of surface quality and environmental impact. A multiobjective optimization approach based on grey relational analysis is applied to optimize the process parameters of the milling process.

This research showed that different levels of machining parameters have different impacts on responses. It should be noted that cutting speed and milling width do not significantly impact the SR or TEI. Feed rate and CT have a dominant impact on surface quality, while milling depth dominantly affects TEI.


Increases in cutting speed lead to fewer environmentally negative impacts (positive effect), while the effect on roughness is small. The reverse holds true for the decrease in cutting speed.Increases in feed contribute to higher SR (negative effect) and fewer environmentally negative impacts (positive effect). A reduced feed contributes to higher SR (negative effect) as well.Increases in milling depth contribute to lower SR (positive effect) and fewer environmentally negative impacts (positive effect). The reverse holds true for the decrease in milling depth.Increases in milling width contribute to higher SR (negative effect) and fewer environmentally negative impacts (positive effect). The reverse holds true for the decrease in milling width.In term of cutting conditions, dry conditions contribute to higher SR (negative effect) and more environmentally negative impacts (negative effect) while in wet conditions this situation is the opposite.


The multiobjective optimization method we proposed demonstrates that a cutting speed of 165 m/min, a feed rate of 0.28 mm/rev, a depth of cut of 2 mm, and a width of cut of 3 mm are the optimal combination of milling parameters. The results of experimental investigation indicated that milling should be performed with a maximal cutting speed, a higher feed rate, a maximal depth, a lower milling width, and wet conditions. This established compromise between the adverse requirements for higher quality and lower environmental impact. The results of multiobjective optimization showed that the solution lies in controlling the cutting parameters. The results can be applied to various machining scenarios. The experimental results reveal that multiple milling performance goals, such as surface quality and environment negative impacts, can simultaneously be optimized by using the multiobjective optimization method we have proposed. The authors of this paper are planning a future study to investigate the melting temperatures or yield strengths of the various materials during milling processes.

## Figures and Tables

**Figure 1 materials-15-08231-f001:**
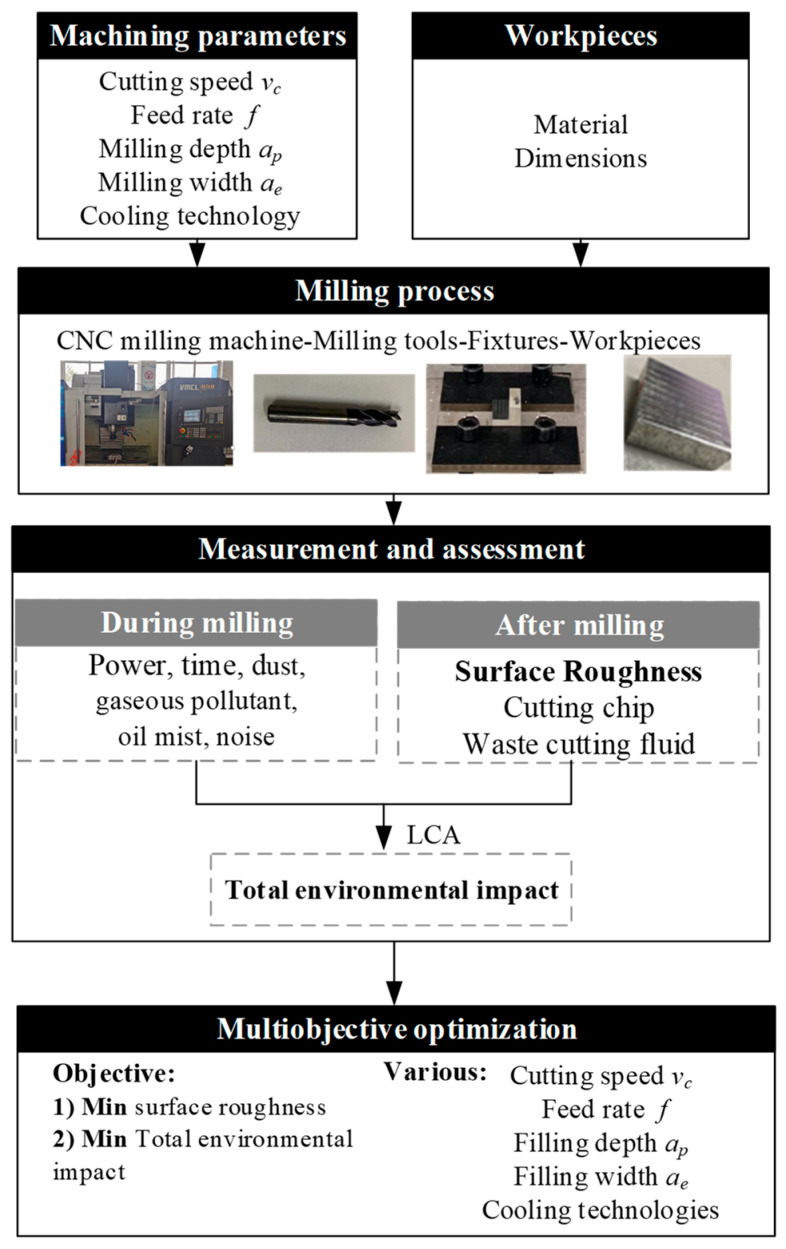
Research methodology.

**Figure 2 materials-15-08231-f002:**
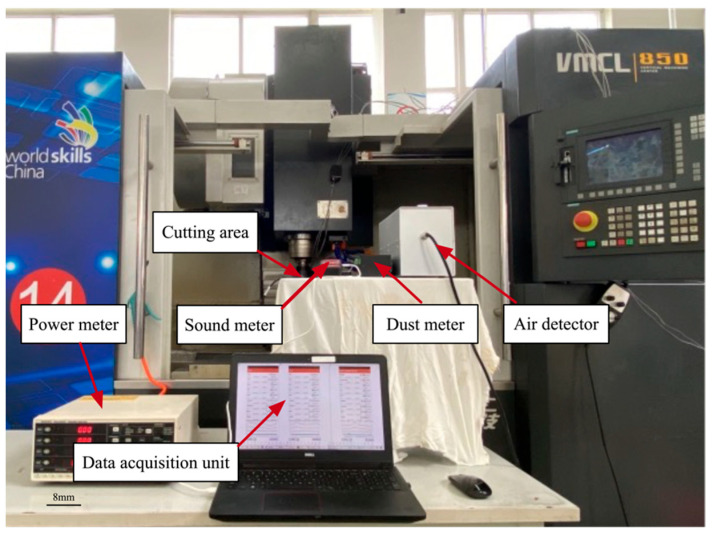
Experimental setup.

**Figure 3 materials-15-08231-f003:**
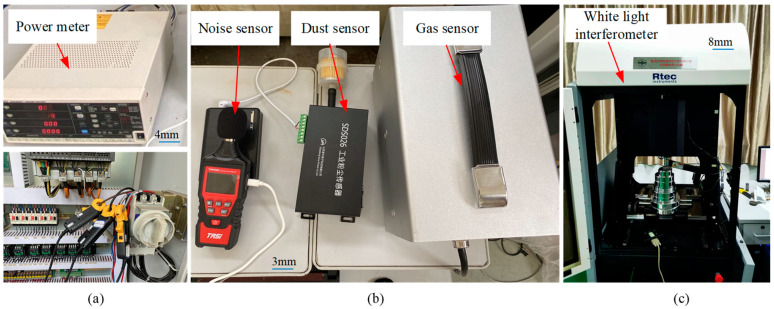
The measuring instrument. (**a**) The power meter; (**b**) the instruments of noise, dust, and air measuring; (**c**) a white light interferometer.

**Figure 4 materials-15-08231-f004:**
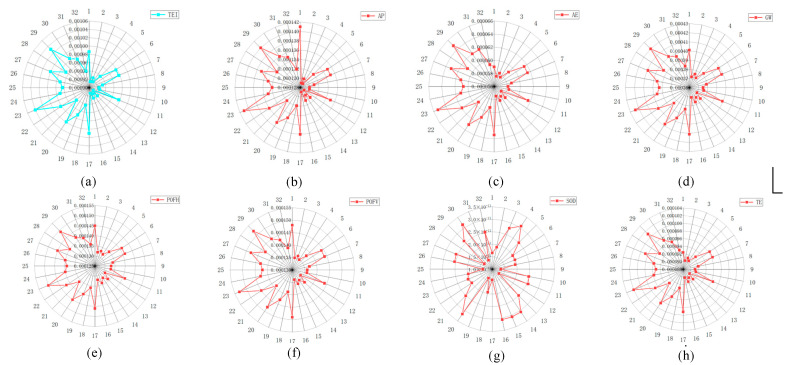
Life cycle assessment results corresponding to the impact categories. (**a**) TEI; (**b**) AP; (**c**) AE; (**d**) GW; (**e**) POFH; (**f**) POFV; (**g**) SOD; (**h**) TE.

**Figure 5 materials-15-08231-f005:**
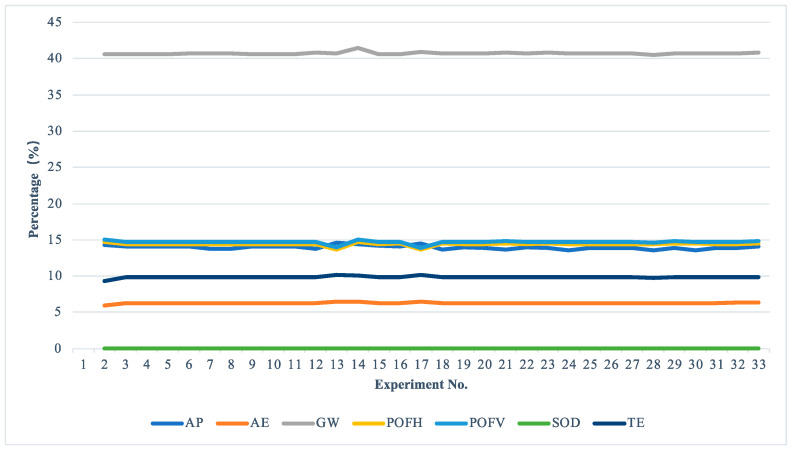
Percentage contributions of the impact categories on the TEI.

**Figure 6 materials-15-08231-f006:**
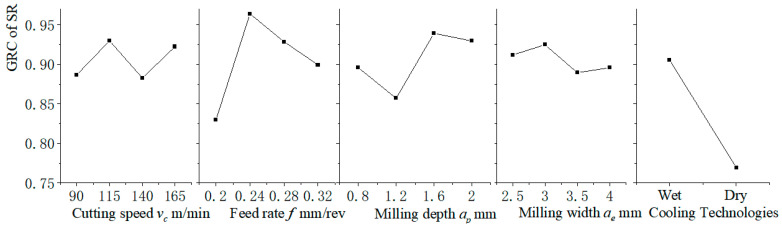
Main effects of SR on GRC for each factor.

**Figure 7 materials-15-08231-f007:**
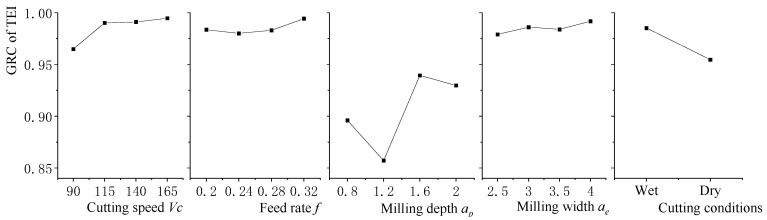
The main effect of TEI on GRC for each factor.

**Figure 8 materials-15-08231-f008:**
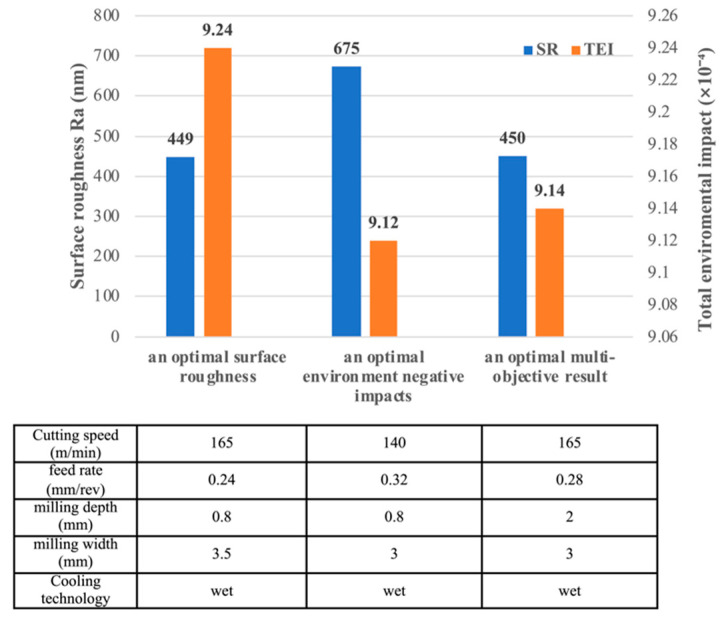
Comparison of optimization results with the different strategies.

**Table 1 materials-15-08231-t001:** An orthogonal array design table based on the set of points and parameters selected for the analysis.

No.	CT	Cutting Speed *v_c_* (m/min)	Feed Rate, *f* (mm/rev)	Milling Depth *a_p_* (mm)	Milling Width *a_e_* (mm)
1	Wet	90	0.2	0.8	2.5
2	Wet	165	0.2	1.2	4
3	Wet	115	0.28	0.8	4
4	Wet	140	0.2	2	3.5
5	Wet	140	0.24	1.6	4
6	Wet	90	0.24	1.2	3
7	Wet	140	0.28	1.2	2.5
8	Wet	115	0.32	1.2	3.5
9	Wet	165	0.24	0.8	3.5
10	Wet	165	0.32	1.6	2.5
11	Wet	90	0.28	1.6	3.5
12	Wet	140	0.32	0.8	3
13	Wet	115	0.2	1.6	3
14	Wet	90	0.32	2	4
15	Wet	115	0.24	2	2.5
16	Wet	165	0.28	2	3
17	Dry	90	0.2	0.8	2.5
18	Dry	165	0.2	1.2	4
19	Dry	115	0.28	0.8	4
20	Dry	140	0.2	2	3.5
21	Dry	140	0.24	1.6	4
22	Dry	90	0.24	1.2	3
23	Dry	140	0.28	1.2	2.5
24	Dry	115	0.32	1.2	3.5
25	Dry	165	0.24	0.8	3.5
26	Dry	165	0.32	1.6	2.5
27	Dry	90	0.28	1.6	3.5
28	Dry	140	0.32	0.8	3
29	Dry	115	0.2	1.6	3
30	Dry	90	0.32	2	4
31	Dry	115	0.24	2	2.5
32	Dry	165	0.28	2	3

**Table 2 materials-15-08231-t002:** The grey-relational coefficient, grade, and associated rank of each experiment.

No.	Grey-Relational Coefficient		Weighted Grey Relational Grade (Greenness)	Rank
	Total Environmental Impact, TEI	Surface Roughness, Ra		
1	0.950863	0.817325	0.884094	21
2	0.997909	0.719831	0.85887	22
3	0.991003	0.94894	0.9699715	6
4	0.996521	0.830416	0.9134685	16
5	0.982167	0.97471	0.9784385	4
6	0.95658	0.931572	0.944076	11
7	0.985547	0.908563	0.947055	10
8	0.991003	0.868614	0.9298085	15
9	0.99169	1	0.995845	2
10	0.990318	0.971889	0.9811035	3
11	0.95658	0.85913	0.907855	18
12	1	0.817692	0.908846	17
13	0.98895	0.952126	0.970538	5
14	0.995828	0.939897	0.9678625	8
15	0.989634	0.950082	0.969858	7
16	0.998605	0.99827	0.9984375	1
17	0.935948	0.771799	0.8538735	23
18	0.978811	0.701895	0.840353	24
19	0.960429	0.820452	0.8904405	20
20	0.942725	0.935598	0.9391615	12
21	0.978811	0.818477	0.898644	19
22	0.953395	0.72025	0.8368225	28
23	0.917949	0.673294	0.7956215	31
24	0.960429	0.632256	0.7963425	30
25	0.96496	0.715146	0.840053	26
26	0.960429	0.941959	0.951194	9
27	0.942105	0.738581	0.840343	25
28	0.963013	0.59748	0.7802465	32
29	0.923871	0.753314	0.8385925	27
30	0.952128	0.680452	0.81629	29
31	0.958501	0.914803	0.936652	14
32	0.980822	0.894875	0.9378485	13

## Data Availability

The raw/processed data required to reproduce these findings cannot be shared at this time as the data also forms part of an ongoing study.

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
