# Peer review of "Evaluation and Improvement of Greenness for Milling AL6061 Alloy through Life Cycle Assessment and Grey Relational Analysis"

_materials, 2022, doi:10.3390/ma15228231_

Round 1

Reviewer 1 Report

Dear Ms. Flora Mao,

Thank you for considering me as a potential reviewer for this article.

In addition to the comments embedded in the pdf (attached along with this review) I have few general concerns.

1. There are several figures that are missing scales or have a lower resolution.

2. Terms like grey relational coefficient (GRC) and grey relational grade (GRG) and overall grey relational analysis (GRA) has been used in this article without providing much of the background. The number of references is not sufficient when it comes to GRA, and the discussion does not necessarily provide enough understanding.

3. I strongly suggest including a list of abbreviation in the beginning of the article.

4. Discussion is inadequate. Discussion section provides conclusions instead of discussion.

5. The whole analysis has been centered on one aluminum alloy. I think the title should be revised. Also, I believe, the results cannot be generalized or extrapolated for other alloys.

6. If, melting temperature or yield strength of the material is to be included in the GRA, perhaps the study could be generalized to an extent.

Please let me know if you have any questions.

Thank you.

Reviewer 2 Report

The paper on Evaluation and improvement of greenness for milling process through life cycle assessment and grey relational analysis, by Z. Xing et al., presents an analysis case of machining under the consideration of a 'greenness' optimization. The Authors used basic machining factors for milling, and add external parameters to evaluate the green component together with the machining quality as provided by the surface roughness. A life-cycle assessment allows for selecting the machining parameters that better suit to different aspects, being the optimization a multi-objective problem.

The paper novelty is limited, but it is interesting as a study case, where sustainability and 'greenness' are addressed beyond simple mechanical parameters to make the best choice of production parameters in a specific process. The analysis provides hints for deploying to many situation that would be useful in production planning, oriented to prioritizing sustainability effects based on a LCA method.

The paper lacks a clear explanation about some points and sections that must be addressed soundly before considering publication. The text is poor in general, with many repetitions and bad constructions that results in confusing parts, and a lengthily text. The language must also be improved through the whole text.

I believe that the Authors can go through the next points and provide a sound revision, as well as read carefully and correct the text to improve the overall quality. An improved version can be considered again, since the analysis is an interesting case and the results may be valuable.

- - - - - - - - - - - COMMENTS - - - - - -

-1- The Authors should provide a draft with line numbers to help in the revision.

-2- There are many sentences in different sections that repeat information again and again. And meaningless sentences. Some are listed in the points below.

-3- The information on sections -2 (Research methodology ) and -3 (Input and process level ) are not well integrated. Indeed, section -2 is poorly exposed. I do strongly recommend that the content of sections 2 and 3 be merged and integrated properly to explain the important parts of the method.

-4- The experimental part should be clear enough as to allow the repetition of the test. On the one side with the details on the test setup. On the other side, specifying the measurement equipment used, including the minimum technical details about the quality of the measurements done (uncertainty). Otherwise, the results are hardly valuable since there is not assessment about their quality.

-5- The situation as explained in Section-5 seems as that the Authors have used a general software that solves the problem by using an internal methodology (GRA based), and so the software makes the job, so the answer must be right. Explain the tools used and how they were probed.

-6- The Authors use the weighting coefficients to define the GRG , without discussion about how they are selected and how hey affect to the results, so the analysis seems arbitrary. Explain clearly how the analysis results depends on these factors. 

-- A detailed revision of more than 15 pages follows. ------------

Reviewer 3 Report

Examining the data, the work here appears to have merit and be of interest. However, the standard of English throughout makes it difficult to read, confusing at points and, at times, difficult to understand. From my interpretation, i think that the content of the paper is logical and scientifically valid, but the ambiguity of the text makes it difficult to know for certain if my interpretation matches what teh author was attempting to convey. As a specific example, the discussion opens by saying:

"The cutting fluid cloud reduces cutting force between the tool, workpiece and chips, the temperature of cutting area is significant decreased and the plastic deformation of metal surface in cutting area is reduced which inhibit the generation of build-up edge."

After several re-reads I interpret this to mean that the lubricating effect of the cutting fluid is reducing the friction and temperature at the cutting site, thus minimizing built-up edge. But the literal, specific wording appears to be implying that aerosolized cutting fluid is counteracting the force applied to the cutting tool, and this is what is minimising the built-up edge. If my most generous interpretation is what the author intended this point makes sense, but if the literal reading of the sentence was the authors intention then it raises many questions i would like to ask them about how they determined this is what is taking place.

The text is filled throughout with these difficult to parse phrases that require a lot of interpretation, and this makes it difficult to be sure if the author is conveying scientifically valid points throughout the article or if I'm simply interpreting valid points that arent actually intended. As such, I think the text would need an extensive rewrite in more clear English before I could properly evaluate its contents and confidently stand behind the results.

Round 2

Reviewer 1 Report

Dear Authors, 

Thank you for making all the necessary changes. 

Author Response

Thank your very much for your carefully review. The English language and style are checked in the revised manuscript.

Reviewer 2 Report

The Reviewer acknowledges the effort done by the Authors to make a thorough revision of the manuscript. Both the presentation and quality are greatly improved now. Still some aspects deserve attention to make a version worth for publication.

Please note that the comment below refers to aspects to improve the text with important information about missing details or necessary information to include in the paper. There are no requirements to extend the experimental results or the analysis done, which is considered from the beginning are rather interesting as to consider the publication.

Please, check the comments below.     

- - - - - - - -

(PAGEx   LINEy)
p4 L9
The AL6061 alloy was in the form of cube having the dimension of 50×50×20 mm3.

The milling process applied should be fully described. The Authors mention a cube. But there is no information about the original workpiece size, or the final part size, or the milling surface or removed material; if the milling is applied to 1 or 5 or 6 faces, etc. This will define the material consumption, for instance.   (note: correct the sentence  : The AL6061 alloy was in the form of a cube having the dimension of 50×50×20 mm3)

The part is small (50*50*20). How do these results apply to larger parts ? Why select this size? The size of the sample is enough to characterize the milling process for the analysis proposed ? The authors should make some comments, even general,  about these points to appreciate if the analysis done is applicable to any milling situation.  Maybe the selection of the study case was done considering an important number of different tests, performed in a limited time and with reasonable conditions to guarantee the quality of the tests measurements.   If this is the 'general reason', please mention it.                     

p5 L7
... lubrication with the speed 0.68g/min. 

Specify how the lubrication was added in the process. Also if it was done 'by hand'.

p5- L11
In this case each range into four points, ... 

correct to: 
In this case, each range was split into four points, as: 

p5 Table -1 

Caption Table 1. Orthogonal array design 

Complete the information as: Orthogonal array design table based on the set of points and parameters selected for the analysis. 

p6  L11
The measuring instrument for measuring Ra called White light interferometer MFT-5000

Note that 'white light interferometry' is a specific technology for non-contact surface roughness measurement. Describe more clearly, kind of:  The instrument for measuring the Ra value MFT-5000 (company details) used the so-called white light interferometry technology to define the surface roughness. 

p7 L9
Gabi 9 software (V8.0)

provide the full reference of the software, and briefly comment its purpose

p7 L18
PETWEU 2004 

Define briefly this and provide reference.

p7 L25
In this study, two objectives of minimizing the SR and TEI are considered   re-write, maybe as: In this study two objectives are considered: minimizing both the SR and the TEI functions. 

p7 L25-ff
GRA is the multi-objective response optimization technique, which developed by Deng in 1989 [28]. GRA commonly used for calculating the performance of variability of complex optimization problems with insufficient data. In the GRA, compare the original sequence and comparable sequence with the help of grey relational grade (GRG). If two sequences are identical, the value of the GRG is equal to 1. The stepwise procedure of GRA is as follow....

The inter-connection of these sentences makes it difficult to read and understand.

Please consider to re-write, maybe as:

GRA is a multi-objective response optimization technique developed by Deng in 1989 [28]. GRA techniques are commonly used for calculating the performance of complex optimization problems with a wide variability and insufficient data. The GRA method evaluates different input sequences with the help of the so-called grey relational grade (GRG) parameter, which allows for the comparison. If two different input sequences result in very close results,  the value of GRG is close to 1, and ideally equal to 1 if identical. The GRA method is applied in several steps, described below. 

p7 L31-ff
The original data have different units. Therefore, it is necessary to transform all dimension into dimensionless index by normalization. Whereas the original data were normalized in the range of 0 and 1, to avoid the effect of response variable units and minimize the variability [33]...

These lines repeat information and are confusing. Consider to re-write it, if this is the meaning, as:  
The original data have different units. To avoid the effect of response variable units and minimize the variability,  the original data were normalized into dimensionless indexes in the range of 0 and 1 [33].  

p8 L13
Takin the experiment ...

Correct to:  
Considering the experiment ...

p8 
Section 5.1 TEI assessment by LCA and figure-4
This subsection deserves a careful reading by the Authors for making a clearer description to the Readers. It should be possible to understand the trends with straight and clear indications, even if the full analysis is more complex and the results are not obvious.
Please, consider the next indications.    

- - - - L13

According to the LCA method in Section 3.4, the LCA results are shown in Figure 4. Takin the experiment No.1, the impact categories AP, AE, GW, POFH, POFV, SOD and TE are 1.41E-4, 5.9E-5, 4.01E-4, 1.45E-4,1.48E-4, 1.65E-11 and 9.24E-5, respectively. And the TEI for this experiment is 9.86E-4.... 

This is a really hard sentence to start the section. Note that the list of values is completely useless in the discussion, and misleading as it is included now in the text.

Consider to re-write as:

The LCA values were evaluated (see Section 3.4) and the results are shown in the charts of the Figure 4 for each of the eight impact categories ( AP, AE, GW, POFH, POFV, SOD and TE, see section 3.4.3) and each according to the 16 different experimental conditions selected in Table-1.

- - - - 
From this point, the Authors should make an effort to show how the chats are useful for the analysis and how results are dependent on some input parameters.    Consider the next presentation sequence: 

1- Comment on only one chart : only one  impact category: what is the maximum and minimum value, and how it evolves through experiments 1-to-16. What is the possible input cause of the maximum and minimum? is there a hint about the key parameter ? Probably it is unclear.

2-  Compare two  impact categories: for what experiments in maximum and minimum? what is the input difference ? is it clear the key parameter? 

Note that if the Authors select carefully one or two cases, it would be possible to understand the general process of the analysis behind the method.

- - - - - - L15 
As shown in Figure 4, the identical trend of negative influence of various machining parameters in all impact categories is depicted ...

How is it possible to see the 'identical trend of negative influence' ? Where ? How? Comparing what?
What is this ? Explain carefully according to the above mentioned points and recommendations, and make it much clearer.  

- - - - L22 & Figure 5
Further, the percentage of each impact category from the TEI for each experiment is shown as Figure 5. It can be seen that the  dominant category of influence is GW (40.57% - 40.90%). All other parameters exert lesser influence, i.e., AP (13.56% - 14.30%), 

Again, what is this ? Explain carefully according to the above mentioned points and recommendations, and  make it much clearer.  

p8 Figure 4
caption
Use the caption to include the minimum information to interpret the figure !

Consider to re-write as: Charts corresponding to the eight impact categories ( AP, AE, GW, POFH, POFV, SOD and TE) and each according to the 16 different experimental conditions selected in Table-1.    

p9 L6
Higher weighted GRG in this paper indicated excellent performance characteristics of the response for SR and TEI...

Correct this sentence.  Consider to re-write it, if this is the meaning, as:  
A higher weighted-GRG value means  better performance characteristics of the response for SR and TEI functions.

p9 L6
...Based on the GRA in Section 4,... 

Correct this sentence.  Consider to re-write as: ...Based on the GRA method (Section 4)... 

p9 L7
...and the rank of each experiment is presented ...
Correct this sentence.  Consider to re-write as:
...and the rank of each experiment, according to the weighted-GRG value, is presented ...

p9 L8
Table 2 shows that the rank of experiment No.16 is 1 and the greenness is highest among all experiments...

Correct this sentence.  Consider to re-write as: 
The experiment No.16 is ranked first, with the maximum weighted-GRG value, therefore being the 'greenness' quality the highest among all the experiments...

p12 Figure 8

- - Caption:  Check the meaning and correct

- -  check the units of SR parameter in the figure

p12 L27
The experiment results show that multiple objectives such as surface quality and 27 environment negative impacts can simultaneously be reduced by using the multi-objective optimization method we proposed. 

Correct this sentence.  Consider to re-write as:
The experimental results show that multiple milling performance goals, such as surface quality and environment negative impacts, can simultaneously be optimized by using the multi-objective optimization method we proposed. 

- - - - - Complete the full company data and the product or model used - - - - - - 

p4 L10 milling machine (TONTEC, China) 
correct to:  milling machine model XXXXXX (Tontec International Ltd., Kowloon Bay, Hong Kong, PRC)

p5 L7 (FUCHS, German) 
correct to:  
specify the lubricant product XXXXXX (FUCHS PETROLUB SE, Mannheim, Germany)

and all  the rest 

p6 L5 (PW3337-02, made by Hioki Electric Co.,Ltd, Range 3W-150kW) 

p6 L7 sound sensor (TASI, 

p6 L7 Dust sensor (Hunan Rike  Instrument Co., LTD 

p6 L9 Air detector instrument (Hunan Rike Instrument Co., LTD, 

p6 L11 MFT-5000 (Rtec, United State) 
